# Comparison of SARS-CoV-2 Antibody Levels after a Third Heterologous and Homologous BNT162b2 Booster Dose

**DOI:** 10.3390/vaccines10101672

**Published:** 2022-10-07

**Authors:** Nesrin Gareayaghi, Mehmet Demirci, Dogukan Ozbey, Ferhat Dasdemir, Harika Oyku Dinc, Ilker Inanc Balkan, Suat Saribas, Neşe Saltoglu, Bekir Kocazeybek

**Affiliations:** 1Blood Center İstanbul Şişli Hamidiye Etfal Training and Research Hospital, 34360 İstanbul, Turkey; 2Department of Medical Microbiology, Medical Faculty, Kırklareli University, 39100 Kırklareli, Turkey; 3Medical Microbiology Department, Faculty of Medicine, Istanbul Okan University, 34959 İstanbul, Turkey; 4Department of Medical Microbiology, Cerrahpaşa Medical Faculty, Istanbul University-Cerrahpaşa, 34098 Istanbul, Turkey; 5Department of Pharmaceutical Microbiology, Faculty of Pharmacy, Bezmialem Vakıf University, 34098 Istanbul, Turkey; 6Department of Infectious Diseases and Clinical Microbiology, Cerrahpaşa Medical Faculty, Istanbul University-Cerrahpaşa, 34098 Istanbul, Turkey

**Keywords:** CoronaVac, BNT162b2, neutralizing antibody, binding antibody

## Abstract

This study aimed to determine the anti-S (receptor binding protein) RBD IgG antibody titers formed against Severe Acute Respiratory Syndrome Coronavirus 2 (SARS-CoV2) and the neutralizing antibody inhibition percentages (nAb IH%) in blood samples taken after two doses of inactive or mRNA-based vaccine and a booster dose. Volunteers with two doses of inactivated CoronaVac (heterologous group; *n* = 75) and BioNTech (BNT)162b2 mRNA vaccine (homologous group; *n* = 75) were included in this study. All participants preferred the BNT162b2 vaccine as a booster dose. First, peripheral blood samples were taken 3 months after the second vaccine dose. Second, peripheral blood samples were taken 1 month after the booster dose. Anti-S-RBD IgG titers were determined by CMIA (SARS-CoV-2 IgG II Quant). Neutralizing antibodies were detected by a surrogate neutralization assay (SARS-CoV-2 NeutraLISA, Euroimmun, Lübeck, Germany). The median age of the volunteers was 40 (IQR 29–47) years old. After the heterologous booster dose, anti-S-RBD IgG levels and neutralizing antibodies increased approximately 50-fold and 9-fold, respectively. Anti-S-RBD IgG titers increased by 9 and 57 times, respectively, while nAb IH% increased by 1.5 and 16 times, respectively, among those with heterologous reminder doses and those with and without a prior history of coronavirus disease (COVID-19). This study showed that after the administration of a heterologous booster dose with BNT162b2 to those whose primary vaccination was with inactivated CoronaVac, the binding and neutralizing antibody levels were similar to those who received a homologous BNT162b2 booster dose. It was observed that the administration of heterologous and homologous booster doses resulted in the development of similar levels of neutralizing antibodies, independently from a prior history of COVID-19.

## 1. Introduction

Due to the appearance of emerging variants of Severe Acute Respiratory Syndrome Coronavirus 2 (SARS-CoV2) all around the world and their spread, the number of COVID-19 cases is rapidly increasing, and because of this, coronavirus disease (COVID-19)-associated morbidity and mortality continue to increase [1]. In our country, vaccination against SARS-CoV-2 with the inactivated CoronaVac vaccine began in July 2020, and then in March 2021, the mRNA-based BioNTech (BNT)162b2 was added as a second choice [2].

Although the efficacy of COVID-19 vaccines is found to be higher than that of vaccines used to prevent other respiratory tract infectious diseases such as influenza, it has been reported that immunity against SARS-CoV-2 wanes, especially after four to six months [3,4]. Understanding antibody-responses-induced post-COVID-19 infection or post-vaccination can guide the improvement of vaccination programs, especially in countries where resources are limited [5]. It is important to detect antibody responses to the virus to understand the effectiveness of current vaccine doses against different and new variants as well as the effect of time on the immune response developed in the host [6]. Vaccines developed with different technologies that have been approved for emergency use by the World Health Organization (WHO) are being used all over the world. Heterologous and homologous vaccinations are used to prevent COVID-19 infections, which are caused by different variants [7].

This study aimed to evaluate immunoglobulin (Ig) G (IgG) antibody titers against the spike protein of SARS-CoV-2 (the anti-S-(receptor binding protein) RBD IgG antibody titers) and the percentage-based inhibition of surrogate neutralizing antibodies (nAb IH%) in individuals admitted to Istanbul University-Cerrahpasa, Cerrahpasa Medical Faculty. These individuals received a booster dose with BNT162b2 after two doses of inactive CoronaVac vaccine (heterologous group) or the BNT162b2 vaccine (homologous group) from peripheral blood samples 3 months after their second dose (CoronaVac or BNT162b2) of vaccine and 1 month after their BNT162b2 booster vaccine. In addition, we aimed to evaluate the relationship between antibody responses and neutralizing antibody capacities after booster dose administration in individuals with and without prior history of COVID-19 in heterologous and homologous groups.

## 2. Materials and Methods

### 2.1. Study Groups and Sampling

Volunteers who had received two doses of the inactivated CoronaVac (Sinovac Life Sciences, Beijing, China) (heterologous group; *n* = 75) vaccine or two doses of the BNT162b2 mRNA (Pfizer-BioNTech) vaccine (homologous group; *n* = 75) were included in this study, which was carried out from 25 April 2022 to 15 August 2022. The two patient groups are not similar in terms of co-morbidities, which could affect immunoresponses (e.g., immunosuppression, diabetes, etc.). All of these participants preferred the BNT162b2 vaccine as a booster dose. Firstly, we collected peripheral blood samples taken 3 months after the second dose of vaccine. Secondly, we collected peripheral blood samples 1 month after the booster dose.

All participants were asked to complete an additional questionnaire regarding their comorbidities and their history of COVID-19 infection. Their COVID-19 histories were confirmed by their medical history (clinical data and/or PCR results). In addition, age, gender, and comorbidity characteristics were comprehensively evaluated. All volunteers provided written informed consent prior to the study.

### 2.2. Evaluation of Anti-S-RBD IgG and nAb IH%

Blood serum was separated from peripheral blood samples at the Istanbul University-Cerrahpasa, Cerrahpaşa Medical Faculty, Serology Unit of the Medical Microbiology Department, and assays were performed at this laboratory. Antibody titers against the S1/RBD region of SARS-CoV-2 were obtained by chemiluminescent microparticle immunoassay (CMIA)-based SARS-CoV-2 IgG II Quant (Abbott, Chicago, IL, USA). The results were obtained as Arbitrary Unit/mL (AU/mL) and were multiplied by the correlation coefficient of 0.142 to convert the results to BAU/mL, which is the reference unit value recommended by the WHO. Neutralizing antibody titers that inhibit the binding of the RBD of SARS-CoV-2/S1 to human angiotensin-converting enzyme 2 (ACE2) were obtained by competitive ELISA-based SARS-CoV-2 NeutraLISA (Euroimmun AG, Lübeck, Germany) assay. The results were obtained as inhibition as a percentage (%IH).

### 2.3. Statistical Analysis

SPSS Statistics v.20.0 (IBM Corp., Armonk, NY, USA) was used to evaluate the data. Qualitative data are presented as numbers and percentages, and quantitative data are presented as median and interquartile range (IQR) 25–75. Qualitative data were evaluated with chi-square χ^2^ and Fisher’s exact test, and quantitative data were evaluated with Student’s t-test and the Mann–Whitney U test. Spearman analysis was used for correlation analysis. Statistical significance was set at *p* < 0.05.

## 3. Results

The demographic data of the 150 volunteers included in the study are presented in Table 1a. In total, 59 volunteers (39.3%) were female, and 91 (60.7%) were male. Thirty-two volunteers (21.3%) had at least one comorbidity. The median age of the volunteers was 40 (IQR 29–47) years old (Table 1b).

The SARS-CoV-2 IgG seropositivity of the participants who received a booster dose of BNT162b2 after two doses of BNT162b2 or two doses of the CoronaVac vaccine are presented in Table 2a. The median anti-S-RBD IgG titers in the first peripheral blood samples, which were taken 3 months after the second dose of BNT162b2, in the homologous group were 7461.3 (IQR 2601.4–14,803.2). In the second sample, which was taken 1 month after the BNT162b2 booster dose, the median of the anti-S-RBD IgG titers was 5331.2 arbitrary units (AU)/mL (IQR 2044–12,253.9). On the other hand, the median anti-S-RBD IgG titers in the first peripheral blood samples, which were taken 3 months after the second dose of CoronaVac, in the heterologous group was 231.8 (IQR 127–471.6). In the second samples, which were taken 1 month after the BNT162b2 booster dose, the median of the anti-S-RBD IgG titers was 10,693.3 AU/mL (IQR 6919.2–17,670.4). While the median nAb IH% of the first samples in the homologous group was 99.24% (IQR 79.33–99.49%), the median nAb IH% of the second samples was 99.14% (IQR 89.55–99.44%). In the heterologous group, the median nAb IH% for the first samples was 10.77% (IQR 2.54–24.57%), and for the second samples, the median was 99.53% (IQR 99.48–99.58%). The anti-S-RBD IgG titers increased approximately 50-fold, and nAb IH% increased approximately 9-fold after the heterologous booster dose (Table 2b).

The antibody results of the participants who received a BNT162b2 booster dose after two doses of BNT162b2 or two doses of the CoronaVac vaccine and who had no prior history of COVID-19 are presented in Table 3. In the first samples of the homologous group, which were taken 3 months after second dose of BNT162b2, the median of the anti-S-RBD IgG titers was 7643.7 AU/mL (IQR 3188.1–14,948.6), and in the second samples, which were taken 1 month after receiving a BNT162b2 booster dose, the median of the anti-S-RBD IgG titers was 3291 AU/mL (IQR 1673–10,420.1). In participants who had no prior history of COVID-19 and who received two doses of the CoronaVac vaccine, the median of the anti-S-RBD IgG antibody titers was 195.9 (IQR 115.28–317.23) in their first samples, which were taken 3 months after the second dose of CoronaVac; in their second samples, which were taken 1 month after receiving a BNT162b2 booster dose, the median was 11,271.2 AU/mL (IQR 6871.8–18,165.03). In the homologous group, the median nAb IH% of their first and second samples was 99.34 (IQR 83.33–99.49%) and 98.28% (IQR 85.34–99.39%), respectively; in the heterologous group, the median nAb IH% of their first and second samples was 6.26% (IQR 1.9–19.27%) and 99.52% (IQR 99.42–99.58%), respectively. After a heterologous booster dose, anti-S-RBD IgG titers increased approximately 57-fold, and nAb IH% increased approximately 16-fold (Table 3).

The antibody results of volunteers who received a BNT162b2 booster dose after two doses of BNT162b2 or two doses of CoronaVac and who had a history of COVID-19 are presented in Table 4. Of the participants in the homologous group, in their first samples, which were taken 3 months after the second dose of BNT162b2, the median of the anti-S-RBD IgG titers was 5560.25 AU/mL (IQR 1905.08–14,662.3), and in their second samples, which were taken 1 month after a homologous booster dose with BNT162b2, the median of the anti-S-RBD IgG titers was 8676 AU/mL (IQR 3872.95–18,322.25). Additionally, in the participants with a history of COVID-19 who received two doses of CoronaVac, the median of the anti-S-RBD IgG antibody titers in their first samples, which were taken 3 months after the second dose of CoronaVac, was 1097.8 (IQR 322.3–2725.25), and in their second samples, which were taken 1 month after a heterologous booster dose with BNT162b2, the median was 9202 AU/mL (IQR 6792.75–13,770.95). In the homologous group, the median nAb IH% of the first and second samples was 96.8 (IQR 62.49–99.45%) and 99.39% (IQR 96.92–99.5%), respectively. In the heterologous group, the median nAb IH% of the first and second samples was 68.7% (IQR 22.35–89.85%) and 99.54% (IQR: 99.51–99.59%), respectively. After a heterologous booster dose, the anti-S-RBD IgG titers increased approximately 9-fold, and nAb IH% increased approximately 1.5-fold (Table 4).

In participants without a history of COVID-19, no correlation was found between anti-S-RBD IgG titers and nAb IH% (*p* > 0.05). On the contrary, in participants with a history of COVID-19, a moderate correlation (r_s_: 0.530, *p* = 0.001) was found between the first anti-S-RBD IgG and nAb IH% levels, and a weak correlation (r_s_: 0.362, *p* = 0.017) was found between the anti-S-RBD IgG and nAb IH% levels in the second samples.

## 4. Conclusions

Booster doses are an important tool for enhancing immunity that tends to wane over time and for enhancing immunity against circulating variants of SARS-CoV-2. Determining antibody levels can be useful for predicting the immune status that will occur after receiving booster doses. In the fight against COVID-19, it is important to obtain data on antibody status after heterologous or homologous booster doses of the different vaccines that are in use after being approved for emergency use [8]. Therefore, in our study, we focused on the status of the anti-S-RBD IgG titers and nAb IH% levels in cases of heterologous and homologous booster dose administration after two doses of the inactive CoronaVac vaccine or two doses of the mRNA-based BNT162b2 vaccine. We found that the levels of anti-S-RBD IgG and neutralizing antibodies increased approximately 50-fold and nine-fold, respectively, after a heterologous booster dose. Anti-S-RBD IgG titers increased by 9 and 57 times, respectively, while nAb IH% increased by 1.5 and 16 times, respectively, among those with heterologous reminder doses and those with and without a prior history of COVID-19.

A study conducted in Italy found that the initial effectiveness of COVID-19 vaccination was 76–92% within 6 months and that it decreased to 34–80% 6 months after vaccination. However, when booster doses are administered, they are reported to reduce SARS-CoV-2 infections by 65%, COVID-19-related hospitalizations by 69%, and deaths by 97% compared to vaccine efficacy 6 months after vaccination [9]. Another study with 4868 participants found that antibody response was significantly reduced 6 months after receiving a second dose of the BNT162b2 vaccine, particularly in men, people older than 65 years of age, and those who are immunocompromised [10]. In our study, we have shown that neutralizing antibody levels increased with booster doses.

Clemens et al. reported that they found low antibody titers after two doses of inactive CoronaVac but that antibody titers and neutralizing antibody levels significantly increased after heterologous or homologous booster doses [11]. Similarly, in our study, anti-S-RBD IgG titers and surrogate neutralizing antibody levels were found to be significantly lower in those who received two doses of the CoronaVac vaccine compared to two doses of the BNT162b2 vaccine. In the group whose primary doses were of inactive CoronaVac, a significant increase was detected in the anti-S-RBD IgG titers and surrogate neutralizing antibody levels in the first month after heterologous BNT162b2 booster dose administration.

Perez-Then et al. reported that the administration of a BNT162b2 booster dose after two doses of inactivated CoronaVac vaccine increased the level of neutralizing antibodies against the Omicron variant of SARS-CoV-2 by approximately 1.5-fold [12]. In our study, it was determined that nAb IH% increased 1.5-fold as a result of heterologous booster vaccination in people who had a history of COVID-19 and 16 times in those without a history of COVID-19. Suah et al. reported that homologous CoronaVac application has low efficiency and that a heterologous booster dose is preferred for the mRNA-based BNT162b2 vaccine [13]. Our study showed that administering a heterologous BNT162b2 booster dose to individuals vaccinated with only two doses of the inactivated CoronaVac vaccine may provide similar efficacy to administering a BNT162b2 booster dose to individuals vaccinated with two doses of the BNT162b2 vaccine.

Liang et al. reported that anti-S-RBD IgG titers increased to 639.30 AU/mL after the administration of a homologous CoronaVac booster dose to individuals who received two doses of the inactivated CoronaVac vaccine. They also reported that the half-life of the neutralizing antibodies and the anti-S-RBD antibodies were 56 days and 82 days, respectively [14]. While the results that we obtained after two doses of inactivated CoronaVac vaccine were similar to those of their study, it was determined that the addition of a heterologous BNT162b2 booster dose increased anti-S-RBD IgG and neutralizing antibody levels to a similar level as homologous BNT162b2 vaccine administration.

Karaba et al. reported a strong correlation between neutralizing antibody levels and anti-S-RBD-IgG levels [15]. In our study, we found a moderate correlation between anti-S-RBD IgG titers and nAb IH% levels in the first blood samples of individuals with a history of COVID-19 infection and a weak correlation between anti-S-RBD IgG titers and nAb IH% levels in the second blood samples. We propose that the difference between studies may be due to the genetic and immune status differences in the patient groups.

Similar to our study, in a Brazilian study using the inactivated CoronaVac vaccine and the BNT162b2 vaccine in combination, high and permanent protection was obtained against the Omicron variant after a heterologous booster dose with the BNT162b2 vaccine [16]. Similarly, in a large cohort study involving approximately 11 million individuals, it was reported that administering homologous or heterologous booster doses to individuals vaccinated with two doses of inactive CoronaVac resulted in a high protection against SARS-CoV-2, including against severe morbidity and death. It was stated that the heterologous booster dose has a higher efficiency than homologous booster doses; therefore, the heterologous booster dose can be applied in clinical practice [17]. When booster doses administered with different vaccines were received after the inactivated CoronaVac vaccine, it was reported that a high T cell response occurs after heterologous booster doses, especially from mRNA-based vaccines (mRNA-1273 and BNT162b2). Similar to our study, it was determined that binding and neutralizing antibody (nAb) levels significantly increased after a heterologous booster dose [18].

The limitations of our study are that it is single-centered, it was not performed on a very large cohort, and we did not check the baseline antibody titers of the participants. The study also uses short-term follow-up data, and it is not known which variant infected those with a history of COVID-19. On the other hand, the data we obtained support vaccination strategies using heterologous booster doses, making them compatible with the literature.

The data of our study show that after the administration of a heterologous booster dose with BNT162b2 to those whose primary vaccination was with the inactivated CoronaVac vaccine, binding and neutralizing antibody (nAb) levels were similar to those who received a homologous BNT162b2 booster dose. While the change in the binding and nAbs was correlated in those who had a previous COVID-19 infection, this correlation was not found in those without a history of COVID-19 infection. It was observed that the administration of heterologous and homologous BNTableT162b2 booster doses resulted in the development of similar levels of neutralizing antibodies, regardless of whether or not the patient had been infected with COVID-19. Although it can be seen that data on heterologous booster dose applications have a positive effect on the host immune response, these data should be supported by more comprehensive and multicenter studies in order to update vaccine policies in regions such as Turkey, where inactivated and mRNA-based vaccines are used together.

## Figures and Tables

**Table 1 vaccines-10-01672-t001:** (a) Distribution of demographic data of all participants. (b) Distribution of age and laboratory data of all participants (median (IQR25/IQR75)).

(a)	
Data	*n* (%)
Gender (F/M)	59 (39.3%)/91 (60.7%)
Comorbidity (no/yes)	118 (78.7%)/32 (21.3%)
Prior COVID-19 infections (no/yes)	107 (71.3%)/43 (28.7%)
**(b)**	
Data	Median (IQR25–IQR75)
Age (median (IQR25/IQR75))	40.00 (29.00–47.00)
First anti-S-RBD IgG AU/mL (median (IQR25/IQR75)) *	1178.85 (204.80–7653.75)
Second anti-S-RBD IgG AU/mL (median (IQR25/IQR75)) *	9080.45 (3694.33–15,230.20)
First nAb IH% (median (IQR25/IQR75)) *	65.60 (9.86–99.30)
Second nAb IH% (median (IQR25/IQR75)) *	99.44 (98.40–99.53)

* First (3 months after two dose); second (1 month after booster dose).

**Table 2 vaccines-10-01672-t002:** (a) Distribution of demographic data of participants who received a booster dose of BNT162b2 after two doses of BNT162b2 or two doses of the CoronaVac vaccine. (b) Distribution of age and antibody results in participants who received a BNT162b2 booster dose after two doses of BNT162b2 or two doses of CoronaVac vaccine (median (IQR25/IQR75)).

Groups	Two Doses BNT162b2 + One BNT162b2 Booster Dose (*n*:75)	Two Doses CoronaVac + One BNT162b2 Booster Dose (*n*:75)	
(**a**)			
Data	*n* (%)	*n* (%)	*p*
Gender (F/M) *n* (%)	9 (12%)/66 (88%)	50 (66.7%)/25 (33.3%)	<0.05
Comorbidity (no/yes) *n* (%)	70 (93.3%)/5 (6.7%)	48 (64%)/27 (36%)	<0.05
Prior COVID-19 infections (no/yes) *n* (%)	49 (65.3%)/26 (34.7%)	58 (77.3%)/17 (22.7%)	0.105
(**b**)			
	Median (IQR25–IQR 75)	Median (IQR25–IQR75)	*p*
Age (median (IQR25/IQR75))	34 (28–42)	43 (34–51)	<0.05
First anti-S-RBD IgG AU/mL *	7461.30 (2601.40–14,803.20)	231.80 (127.00–471.60)	<0.05
Second anti-S-RBD IgG AU/mL **	5331.20 (2044.00–12,253.90)	10,696.30 (6919.20–17,670.40)	<0.05
First nAb IH% *	99.24 (79.33–99.49)	10.77 (2.54–24.57)	<0.05
Second nAb IH% **	99.14 (89.55–99.44)	99.53 (99.48–99.58)	<0.05

* First (3 month after two dose vaccination), ** Second (1 month after third booster vaccination dose).

**Table 3 vaccines-10-01672-t003:** Distribution of antibody results in subjects without prior history of COVID-19 who received a BNT162b2 booster dose after two doses of BNT162b2 or two doses of CoronaVac vaccine.

Groups	Two Dose BNT162b2 + One BNT162b2 Booster Dose (*n*:49)	Two Dose CoronaVac + One BNT162b2 Booster Dose (*n*:58)	
	Median (IQR25–IQR75)	Median (IQR25–IQR75)	*p*
Age (median (IQR25/IQR75))	33 (27–41.5)	43 (32–51)	<0.05
First anti-S-RBD IgG AU/mL *	7643.70 (3188.10–14,948.60)	195.90 (115.28–317.23)	<0.05
Second anti-S-RBD IgG AU/Ml **	3291.00 (1673.00–10,420.10)	11,271.20 (6871,80–18,165.03)	<0.05
First nAb IH% *	99.34 (83.33–99.49)	6.26 (1.90–19.27)	<0.05
Second nAb IH% **	98.28 (85.34–99.39)	99.52 (99.42–99.58)	<0.05

* First (3 month after two dose vaccination), ** Second (1 month after third booster vaccination dose).

**Table 4 vaccines-10-01672-t004:** Distribution of antibody results in subjects with a history of COVID-19 who received a BNT162b2 booster dose after two doses of the BNT162b2 or two doses of the CoronaVac vaccine.

Groups	Two Dose BNT162b2 + One BNT162b2 Booster Dose (*n*:26)	Two Dose CoronaVac + One BNT162b2 Booster Dose (*n*:17)	
	Median (IQR25–IQR75)	Median (IQR25–IQR75)	*p*
Age (Median (IQR25/IQR75))	37 (31.75–43)	44 (34–52)	<0.05
First anti-S-RBD IgG AU/mL *	5560.25 (1905.08–14,662.30)	1097.80 (322.30–2725.25)	<0.05
Second anti-S-RBD IgG AU/mL **	8676.50 (3872.95–18,322.25)	9202.00 (6792.75–13,770.95)	0.585
First nAb IH% *	96.80 (62.49–99.45)	68.70 (22.35–89.85)	<0.05
Second nAb IH% **	99.39 (96.92–99.50)	99.54 (99.51–99.59)	<0.05

* First (3 month after two dose vaccination), ** Second (1 month after third booster vaccination dose).

## Data Availability

The data that support the findings of this study are available from the corresponding author upon reasonable request.

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
