# Peer review of "Comparison of SARS-CoV-2 Antibody Levels after a Third Heterologous and Homologous BNT162b2 Booster Dose"

_vaccines, 2022, doi:10.3390/vaccines10101672_

Round 1

Reviewer 1 Report

Useful information

Author Response

Reply to the Review Report (Reviewer 1)

  1. Moderate English changes required.
  2. English editing was done by MDPI editing services. The editing certificate was attached.

Reviewer 2 Report

This is an interesting and very informative study for the treating physicians, as the issue of heterologous vaccination schemes arises in several countries. However a few things need to be addressed

a. the data have to be better and more sharply presented in the manuscript. The tables are difficult to follow; the two groups are being compared both between columns and between lines. How come a patient group is being compared both with first and second samples?

b. natural immunity is known to confer better immunity and higher levels of neutralizing antibodies. How do the authors explain the fact that heterologous vaccination confers higher neutralizing antibodies than natural immunity?

c. the two patient groups are not similar in terms of co-morbidities which as known may affect immunoresponse (eg immunosuppression, diabetes etc). This has to be stressed out

d. all abbreviations need to be explained

Author Response

Reply to the Review Report (Reviewer 2)

  1. the data have to be better and more sharply presented in the manuscript. The tables are difficult to follow; the two groups are being compared both between columns and between lines. How come a patient group is being compared both with first and second samples?
  2. We re-arranged all tables. We divide table 1 and 2 as Table 1a, 1b and Table 2a and Table 2b. We showed one column in each group like female (n, %), prior COVID-19 infection (n, %), age (medium, IQR). Actually, we compared only the two patient groups and obtained a p” value. We did not compare the first and second samples of the same patient group. We only showed how many times the antibody levels increased.
  3. natural immunity is known to confer better immunity and higher levels of neutralizing antibodies. How do the authors explain the fact that heterologous vaccination confers higher neutralizing antibodies than natural immunity?
  4. We couldn’t know this, because we did not measure the baseline neutralizing antibody levels after the natural infection and added the following sentence “The limitations of our study are that it is single-centered, it was not performed on a very large cohort, and we did not check the baseline antibody titers of the participants.” as the limitation of the study to the Discussion section of the manuscript.
  5. the two patient groups are not similar in terms of co-morbidities which as known may affect immunoresponse (eg immunosuppression, diabetes etc). This has to be stressed out
  6. Thanks for this comment and added “The two patient groups are not similar in terms of co-morbidities which as known may affect immunoresponse (eg immunosuppression, diabetes etc). To the Material and Methods section 2.1 Study groups and sampling.
  7. all abbreviations need to be explained
  8. Some missing ones in the previous version was explained in their first use.

- the anti-S- (receptor binding protein) RBD IgG antibody titers

- Severe Acute Respiratory Syndrome Coronavirus 2 (SARS-CoV2)

- BioNTech (BNT)162b2

- Coronavirus disease (COVID-19)

- arbitrary units (AU)/mL

Reviewer 3 Report

This study assessed the anti-S-RBD IgG antibody titers formed against SARS-CoV-2 and the neutralizing antibody inhibition percentages (nAb IH%) in blood samples taken after two doses of inactive or mRNA-based vaccine and the booster dose. It was found that after heterologous booster dose, anti-S-RBD IgG levels and neutralizing antibodies increased approximately 50- fold and 9-fold, respectively. Anti-S-RBD IgG titers increased by 9 and 57 times, respectively, while nAb IH% increased by 1.5 and 16 times, respectively, among those with heterologous reminder doses and those with and without prior history of COVID-19. Overall, the study is well-designed and the manuscript is well-written. Therefore, I just have several minor suggestions.

1.      Please mention the study period in the method section.

2.      Please revise the first two paragraph in the discussion and stated your main findings first.

3.      The tables are difficult to read, so please show one column in each group like female (n, %), prior COVID-19 infection (n, %), age (medium, IQR)

Author Response

Author's Reply to the Review Report (Reviewer 3)

  1. Please mention the study period in the method section.

-   The following sentence was added to the study groups “this study which was carried out from 25.04.2022 to 15.08.2022.

  1. Please revise the first two paragraphs in the discussion and stated your main findings first.

-    We revised the first two paragraphs of the discussion and added “We found that the levels of anti-S-RBD IgG and neutralizing antibodies increased approximately 50- fold and nine-fold, respectively, after a heterologous booster dose. An-ti-S-RBD IgG titers were increased by 9 and 57 times, respectively, while nAb IH% increased by 1.5 and 16 times, respectively, among those with heterologous reminder doses and those with and without a prior history of COVID-19.” to the end of first paragraph of the discussion as our main findings.

  1. The tables are difficult to read, so please show one column in each group like female (n, %), prior COVID-19 infection (n, %), age (medium, IQR).

-    We re-arranged all tables. We divide table 1 and 2 as Table 1a, 1b and Table 2a and Table 2b. We showed one column in each group like female (n, %), prior COVID-19 infection (n, %), age (medium, IQR).